# Virtual Modeling of User Populations and Formative Design Parameters

**Benjamin M. Knisely and Monifa Vaughn-Cooke ***

Department of Mechanical Engineering, University of Maryland, College Park, MD 20742, USA;
bknisely@terpmail.umd.edu
* Correspondence: mvc@umd.edu

**Abstract:** Human variability related to physical, cognitive, socio-demographic, and other factors can contribute to large differences in human performance. Quantifying population heterogeneity can be useful for designers wishing to evaluate design parameters such that a system design is robust to this variability. Comprehensively integrating human variability in the design process poses many challenges, such as limited access to a statistically representative population and limited data collection resources. This paper discusses two virtual population modeling approaches intended to be performed prior to in-person design validation studies to minimize these challenges by: (1) targeting recruitment of representative population strata and (2) reducing the candidate design parameters being validated in the target population. The first approach suggests the use of digital human models, virtual representations of humans that can simulate system interaction to eliminate candidate design parameters. The second approach suggests the use of existing human databases to identify relevant human characteristics for representative recruitment strata in subsequent studies. Two case studies are presented to demonstrate each approach, and the benefits and limitations of each are discussed. This paper demonstrates the benefit of modeling prior to conducting in-person human performance studies to minimize resource burden, which has significant implications on early design stages.

**Keywords:** human variability; human performance; digital human modeling; heterogeneity; design validation

---

## 1. Introduction

Human performance in complex systems can have a significant influence on overall system performance [1]. Models of human performance can aid designers in predicting human behavior in response to design decisions, which presents the opportunity to evaluate designs without expensive physical prototyping and testing. Additionally, these models can provide a quantitative framework to optimize system design for performance, reliability, and safety [2].

Designing a system to maximize human performance requires designers to not just consider the typical user, but also the variability of users. Human variability can have a significant impact on overall user-product interaction by increasing the variability of potential use-cases. Examples of research seeking to quantify human variability in the context of design includes examining manual tool use [3], ear-bud design [4], and smart phone use [5]. Designing for human variability becomes especially important for safety critical systems with highly heterogeneous user populations, such as medical device users, where a single error or suboptimal performance can lead to devesting consequences [6,7].

A critical step in modeling human performance for design validation is retrieving human performance data [2]. Typically, this is done through engaging the population directly. A representative sample of the population is recruited and observed interacting with the system being designed. Design parameters can be varied to examine their influence on human performance, which can be used as input

for modeling. This can be prohibitively expensive and time consuming, especially when the population is highly heterogeneous or difficult to access. Narrowing in on a small set of design parameters prior to engaging the population can reduce the burden of the process. Additionally, identifying appropriate human parameters to stratify the population into meaningful segments can reduce the risk of excluding prominent users.

In this paper, two formative analysis modeling tools are discussed to reduce product and human parameters in the early stages of design prior to human engagement in later design stages. Digital human models (DHMs), 3D virtual representations of humans that can perform simulated interaction with a product [8], can be used to evaluate and reduce design parameters for a human study by simulating performance. Further, population modeling with existing human databases can be used to identify important user characteristics and identify appropriate strata for recruitment in subsequent human performance studies, thereby focusing recruitment efforts and reducing necessary resources. Neither of these tools require subjects to be engaged directly, which can be cost prohibitive for populations who are difficult to access in sufficient numbers. In addition to the long-standing challenges of human data collection and design studies, further complications have been more recently posed by COVID-19, where designers and those conducting studies must reduce direct contact with subjects to adhere to pandemic safety protocols. While these tools don't replace engagement with the target population, they do minimize requirements to engage directly.

First, traditional methods to quantify human performance are discussed, as well as the limitations when working with highly heterogeneous, non-general populations. Next, alternative approaches to quantify human performance and human variability are discussed. Finally, two medical device case studies are used to demonstrate these approaches.

## 1.1. Traditional Approaches to Quantify Human Performance

The most common and direct approach to quantify human performance is via direct observation, typically in a controlled setting. Representative participants are asked to perform the task of interest in the actual context or a simulated environment. Typically, direct objective measurements of performance are utilized, including task accuracy, timing, and failure rates [9–11]. Neurophysiological data, such as pupillometry [12,13] and heart rate data [14], can be used as an indirect measure of human performance by serving as a measure of cognitive workload. Cognitive workload is a multidimensional construct used to represent the effort required by human working memory [15]. Suboptimal workload has been linked to higher failure rates and poor performance [16].

Subjective measures of human performance can also be used in conjunction with direct observation (objective measures) to provide additional robustness, or in substitute when direct observation is not possible. While generally less difficult to apply compared to a controlled study, subjective data suffers from biases inherent to self-reported data [17,18]. Further, surveys and interviews rarely provide measures of performance that can be directly integrated into existing models. Instead, participants usually provide ordinal rankings or qualitative descriptions of their performance which must be converted manually into a quantitative format [19]. Little exact guidance exists on this practice, making researcher experience in the practice critical [20].

The advantage of these traditional approaches is that task performance can be measured directly or indirectly through engagement with a sample of the user population. Further, these studies can be controlled such that the observed performance is minimally influenced by confounding factors. The drawback is that these studies can be time consuming and expensive to perform properly [19]. Designers report many difficulties engaging with end-user populations. Often, organizational culture can prevent appropriate engagement by discouraging the practice as not useful, adhering to preconceived assumptions about the end-users, and having limited knowledge about or access to tools for user engagement. Additionally, design teams may only have access to the project client, who is usually not the end-user and who may not work directly with the population [21].

Finally, for product development teams without human factors and modeling expertise, effectively engaging with users and quantifying their performance may require soliciting outside help.

Further, some populations may be difficult to access regardless of experience or organizational factors. For certain products, such as medical products and devices, vulnerable and minority populations may constitute a disproportionally large segment of the intended user. These populations include (but are not limited to) racial/ethnic minorities, the elderly, low socio-economic status, and those living in rural areas [22]. Insufficient recruitment of these populations in research is common due to a variety of barriers. For example, recruitment of elderly participants for clinical studies is often insufficient due to mistrust in institutions, transportation issues, medical concerns, indifference, sensory and cognitive limitations, and physical limitations [23]. Furthermore, these populations often have capabilities and usability needs not common to the general populations [24]. Thus, human performance data collected from the general population is likely inadequate.

When planning a study where human performance will be measured in response to varying design parameters, minimizing the number of parameters can reduce study duration and cost. This is especially true for heterogeneous populations where large numbers of participants are required. In the next section, digital human models (DHMs) are introduced as a means for reducing design parameters.

### 1.2. Digital Human Modeling

DHMs are virtual representations of humans that can be placed into a simulation or virtual environment to simulate interaction with a product or system [8]. There are many commercially available and open-access software tools available for this type of modeling geared towards product design, which is the primary focus of this section. In product design, these modeling tools are typically used in the early stages of design to evaluate the ergonomic impact (e.g., musculoskeletal effort, user comfort, product fit) of various design decisions [25,26]. Existing modeling software is primarily focused on physical and biomechanical human behavior [27]. There are few tools available for simulation of cognitive tasks and perception, excluding some elements of vision [8]. Measuring performance in terms of accuracy or error is difficult without integration of cognitive models. Performance is primarily focused on biomechanics, physical workload and stress, and product fit.

Software focusing on physical product interaction loosely fall into two camps: simple biomechanical models that are easy to apply and complex biomechanical models where application is more demanding. Many of the software tools with simpler biomechanical models typically treat the human body as a simple system of links and joints. This software excels at evaluating spatial accommodation and clearance/interference of body parts with product components [28]. Many of these contain relatively simple stress/strain calculations as well, such as the static strength prediction and lower back analysis tools in Jack [29]. Popular software falling under this category includes Siemens Jack, Anybody, Ramsis, Catia/Delma Human Model, and Santos [30]. Limited dynamical modeling is possible in most of these software, however musculoskeletal models are generally rather simplistic or non-existent, which can produce unrealistic movements.

Software on the side of more biomechanical complexity is geared towards research applications but may also have use in product development. This software is generally more difficult to apply and requires additional domain knowledge, however, it allows more precise predictions of forces and stress undergone by the body as a result of some action. Perhaps the most well-known software is OpenSim, an open-source biomechanical modeling software [31]. OpenSim models are equipped with state-of-the-art musculoskeletal models which allows them to perform realistic inverse and forward dynamics simulations.

Of the sensory and perceptual modalities, vision analysis is most commonly available with DHMs. Most software capabilities are limited to field of view analysis. For example, Jack provides the use of "vision cones" that demonstrate the theoretical field of view for a human model [32]. Limited cognitive modeling software exists as well that attempt to simulate human decision-making and mental processing in response to stimuli [27]. For example, the Man-Machine Integration Design and Analysis

System (MIDAS) is a dynamic simulation scenario builder developed by NASA Ames Research center that includes cognitive workload models based on the visual, auditory, cognitive, and psychomotor (VACP) workload model [33].

The obvious advantage of these tools when evaluating the impact of human variability is that live participants are not necessary. The burden of recruitment is no longer an issue. DHMs can be used to simulate interaction, primarily physical, and facilitate narrowing down a set of design parameters prior to engaging users. Several design candidates can be developed using CAD and inserted into a DHM environment. Candidates can be varied along design parameters, with a candidate being developed to represent each combinatorial option of the design parameters being varied. A simulated interaction with the virtual models can be used to narrow down design parameters based on interaction performance.

In addition to design parameters, narrowing in on a set of human parameters could be useful for identifying recruitment strata prior to a live human study. Doing so requires identifying the appropriate human parameters relevant to the task of interest, obtaining sample data for these parameters, and modeling the population of interest using this data.

### 1.3. Modeling Variability with Existing Human Data

Human beings have widely varying characteristics (physical, cognitive, demographic, preference, etc.), and these characteristics can have a significant impact on design interaction. Understanding these characteristics for a study population can be beneficial for appropriate representation. In highly heterogeneous populations, representing every user may be difficult, and may require prioritization of highly prominent user types.

Stratification is the process of dividing a study population into distinct sub-populations. Stratification is usually necessary because representing every combination of user characteristic in a study is likely infeasible due to resource burden. When defining strata for the purpose of modeling human variability, one should identify the key user characteristics linked to performance [34]. This will guide the stratification of users for the quantification task. Key characteristics are highly dependent on the system or task being studied and can result in an overwhelming array of characteristics to consider. A highly heterogeneous user population will lead to more complex strata, and thus a more complex representation process. Prioritization of key characteristics will likely be necessary, which can be difficult a priori without input from domain experts or guidance from existing data. Modeling human variability for the purpose of defining strata prior to engaging a population could overcome some of the discussed limitations.

Many publicly available surveys exist with data on human health, capability, and other characteristics. This data can be used to define population characteristics that directly impact product or system interaction. Sources of data include the National Health and Nutrition Examination Survey (NHANES) [35], the National Health Interview Survey [36], and the US Army Anthropometric Survey (ANSUR) [37]. This data can potentially be used to better understand a subject population prior to engaging them. For example, Harrington et al. [38] used NHANES 2009–2010 to stratify the US population by demographic variables to investigate sitting behavior. Kudsk et al., [39] used NHANES 1988–1994 data to assess poor outcomes in hospitalized patients for demographic strata.

As a further example, design for human variability (DfHV) is a design research area that has made extensive use of existing human data for modeling human variability, primarily limited to anthropometric data. Design for human variability is a Design for X approach to user-centered design that focuses on quantifiable, physical characteristics of the user [4]. Utilizing existing anthropometric databases, such as ANSUR [37], users can be virtually fit to a product to assess how well they are spatially accommodated [40]. Statistical modeling can be performed to infer missing data, model correlation between human parameters, and extend the usefulness of the data in practice [41–43]. Often, anthropometric data is accompanied by demographic data, which allows modeling for specific user populations, assuming the demographics of the target population are known [44].

In addition to databases, generating virtual populations of users using summary statistics has also been proposed as a means to analyze user variability [45]. These efforts are limited because they focus on the variability of the human and not the variability of human performance. Certain assumptions must be made about the relationship between the obtained user characteristics and product use to predict human performance. Models or guidance may exist that link user and product characteristics to performance of a task, but the context specific nature of human performance in design makes perfectly transferring these links difficult. Accurately making these assumptions may be difficult for product designers without human factors experience and may be impossible if the exact context (user and product, together) is not well studied.

## 2. Materials and Methods

Section 1 discussed traditional approaches to quantify human performance through direct engagement with subjects (objective and subjective data collection). Two virtual modeling approaches (DHM, population modeling) were introduced that could be used to reduce the complexity of a human performance study prior to direct engagement. These approaches are especially important when studying highly heterogeneous and difficult-to-access populations due to resource constraints.

In this section, the virtual approaches are discussed in further detail in the context of highly heterogenous user populations. A brief design case study will be presented for each. First, DHMs will be discussed as a means to reduce design parameters prior to engaging live subjects. Then, population modeling via existing databases will be discussed as a means for population stratification.

The case studies will both be applied within the medical device domain. The consensus Human Factors Engineering/Usability Engineering guidelines across international regulatory agencies suggest medical device manufacturers perform rigorous human factors and usability testing to minimize the likelihood of unsafe device use. Further, in formative design verification, medical device manufacturers are encouraged to evaluate participants who are "representative of the range of characteristics within their user group," with each group representing distinct user populations who will "perform different tasks or will have different knowledge, experience or expertise that could affect their interactions with elements of the user interface" [46,47]. Thus, medical device manufactures have stake in tools for understanding population variability in the context of their device. Medical device users, particularly for patient-facing devices, are highly heterogeneous. This makes them difficult to access from the manufacturer's perspective, which makes them a suitable subject for these case studies.

### 2.1. Case Study 1: Using DHM to Reduce Product Parameters

DHMs present the opportunity to simulate user device interaction virtually prior to engaging with a population. For this case study, Jack will be used to demonstrate how DHMs can be used to formatively evaluate human performance variability and eliminate potential design parameters. In Jack, CAD models can be inserted into a virtual environment, and a human model can be positioned around the CAD model. Human models are defined with joints and links corresponding to major body segments. To simulate realistic poses, constraints can be placed between points on the human model and points on the CAD model. Jack uses inverse kinematics to manipulate the unconstrained joints and segments while maintaining the constraint relationships.

Jack was used to study different case dimensions for a handheld medical device used by chronic disease patients (diabetes, hypertension, etc.). Examples of these devices are glucometers and blood pressure monitors, which are commonly designed as handheld devices requiring precise manual manipulation and grasping with the hand. Due to common comorbidities such as arthritis, these chronic disease populations may experience difficulties with these devices that the general population do not experience as commonly. This scenario is an ideal use of DHM because of the physical nature of the limitation, and the challenges of directly accessing these populations as discussed earlier.

The goal of this study was to eliminate handheld product dimensions for a future usability study. There is no existing guidance that focuses specifically on chronic disease populations. Most existing

similar work focuses on phone design [48], for which the user population is different than the medical device user population. Therefore, a wide range of candidate dimensions (width, length, etc.) needed to be evaluated, which provides an extensive list of combinatorial dimension options, which are infeasible to explore in the format of an in-person design validation study.

The interaction to be simulated was a user grasping the device with the thumb in a resting position (Figure 1). This interaction requires evaluation because of the prevalence of joint restrictive comorbidities such as arthritis [49,50]. Jack's disembodied hand module was used to simulate interaction with the device. The disembodied hand module is a replica hand where each finger and thumb are comprised of 3 segments. The fingers have 3 degrees of freedom and the thumb has 4.

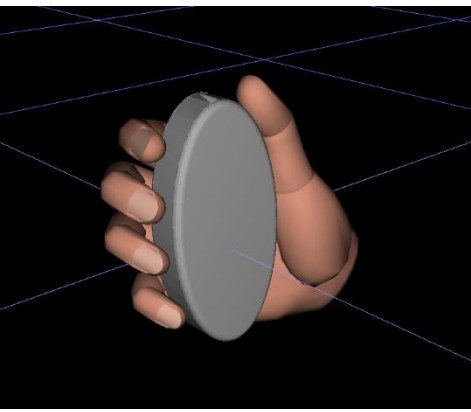

**Figure 1.** Jack disembodied hand posed with Case 8 for the discussed simulation.

As a demonstration, an arbitrary range of hand dimensions were selected to represent a population that does not conform to the general population. The 95th and 5th percentile female hands from this range were positioned with the device. Using boundary "manikins" is common practice to simulate the extreme use cases in human variability modeling, especially when the relationship between user size and device interaction is assumed to be linear [51]. If the relationship was thought to be "U" shaped or some other nonlinear relationship, then it may be desirable to include some sizes in between the 5th and 95th percentile. For the 5th percentile, hand length was 16.5 cm and hand breadth was 7.2 cm. For the 95th percentile, hand length was 18.3 cm and hand breadth was 8.0 cm.

The shape for all cases was an extruded ellipse, which can be defined by length, breadth, and width. The design parameters to be evaluated were case length and case breadth, two dimensions that impact grasp comfort. Three dimensions were selected for each parameter to demonstrate the varying impact of design specifications on a performance outcome. The 3 case lengths were 10.5, 11.0, and 11.5 cm. The 3 case breadths were 3.5, 4.5, and 5.5 cm. Since case length and breadth are assumed to be dependently related for grasping tasks, combinatorial options were generated. The combination of each dimension led to 9 total case designs, listed in Table 1. The width of all cases was 1.75 cm.

**Table 1.** Design parameters for handheld medical device.

|  | Case 1 | Case 2 | Case 3 | Case 4 | Case 5 | Case 6 | Case 7 | Case 8 | Case 9 |
|---|---|---|---|---|---|---|---|---|---|
| Length (cm) | 10.5 | 11.0 | 11.5 | 10.5 | 11.0 | 11.5 | 10.5 | 11.0 | 11.5 |
| Breath (cm) | 3.5 | 3.5 | 3.5 | 4.5 | 4.5 | 4.5 | 5.5 | 5.5 | 5.5 |

The measures of performance for the study were the 3 joint angles associated with the 4 fingers in Jack's hand model. Starting with the joint furthest from the hand, the joints are the distal interphalangeal joint (DIP), the proximal interphalangeal joint (PIP), and the metacarpophalangeal (MCP) joint. All joint angles are measured at a local axis on the joint and measured as deviation from the neutral position.

Jack defines the neutral position for each finger as shown in Table 2. Figure 2 displays the hand in the neutral position.

**Table 2.** Neutral hand position finger joint angles from local joint axis as defined by Jack.

|           | Index | Middle | Ring | Pinky |
| --------- | ----- | ------ | ---- | ----- |
| DIP (deg) | 5.0   | 5.0    | 5.0  | 0.0   |
| PIP (deg) | 15.5  | 15.5   | 15.8 | 11.8  |
| MCP (deg) | 7.0   | 8.0    | 8.0  | 8.0   |

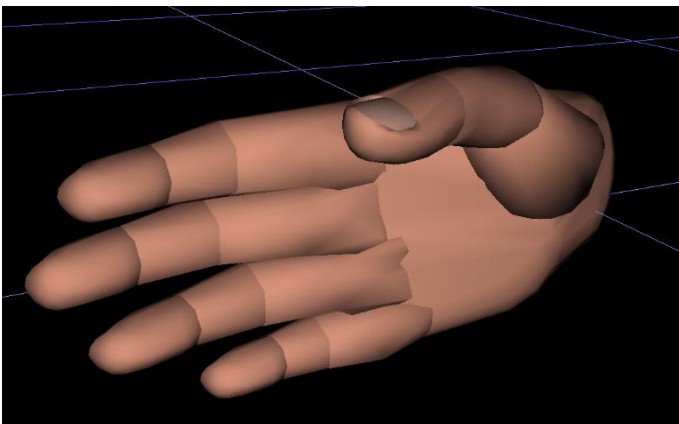

**Figure 2.** Jack disembodied hand in the neutral position.

For handle design, one study demonstrated that subjective comfort decreased on average as object diameter increased, starting at 3.5 cm [52]. For the purposes of this demonstration, we assume that comfort will increase as grip diameter decreases. It thus follows that a preferable case design will have larger finger joint deviations, for the range of cases being evaluated. In reality, there is likely a cut-off point or threshold value where the grip size-comfort relationship reverses, however, this is not considered in this demonstration. Another assumption was made that 1 degree of movement for each joint was equal with respect to ergonomic performance. Therefore, each device casing can be evaluated by the sum of each joint's absolute deviation from neutral posture.

To simulate the interaction, each hand was positioned behind the device, using a central point on the back of the device and on the palm of the hand to ensure device placement was equivalent for each hand size. While several grasping postures exist, the posture used was based on a grasp where all 4 fingers are positioned on the side of the device in a standard location, and the device is braced against the palm [53]. The thumb was not considered relevant for this grasp. Skin compression is not an available feature in Jack so contact between device and hand was made surface to surface. The compressibility of human tissue may influence joint deviation in actual practice. The location of the hand was constrained to a position in global space. The inverse kinematic solver moved the joints to accommodate the constraints, and the resulting joint angles were recorded. Scaling and posing of the hand and extraction of data was automated using JackScript, a Jack module that interfaces with Python.

To further quantify the influence of design parameters on joint deviation, the relationships can be statistically modeled. A linear model was fit to the data, where total joint deviation for each finger was treated individually as a response variable. Case breadth, case length, finger, and hand size (5th or 95th) were included as effects. Breadth and length were treated as categorical.

*2.2. Case Study 2: Using Existing Data to Identify Recruitment Strata*

In addition to evaluating design parameters for the purpose of eliminating options prior to a human study, modeling can be used to better understand the user population. Understanding the user population prior to recruitment can reduce the risk of poorly representing appropriate population heterogeneity or missing key user groups. In this case study, the use of existing human data is used to demonstrate how this can be avoided.

In this paper, we demonstrate how national databases can be used to identify key user group strata, based on variables related to product interaction. The National Health and Nutrition Examination Survey (NHANES) is a longitudinal survey used to study the health and wellbeing of United States citizens. Included are demographic, socioeconomic, dietary, and health-related questions [35]. This data is particularly useful for characterizing user groups because of the many physical and cognitive health characteristics included. For this case study, the hypertensive population will be modeled as the characteristics that are relevant for medical device interaction and impact human performance.

Hypertension is a highly prevalent disease and remains one of the leading causes of death worldwide [54]. This has rendered the blood pressure monitor one of the most commonly used medical devices in the home, yet the ability of the user to appropriately interact with this device is often overlooked in the design process [55]. Users of these devices are highly heterogenous because of their overall prevalence and the diverse physical and cognitive actions required for device interaction. Understanding the capabilities and characteristics of this user population is critical for promoting prolonged adherence. Hypertensive blood pressure monitor users will be the subject of the case study.

The core tasks involved with operating a blood pressure monitor are as follows: (1) Recall of device operating procedures; (2) Fine hand manipulation for assembly and operation of device; (3) Visual detection and perception of device output; (4) Evaluation of device output; and (5) Decision-making regarding subsequent actions. Using these tasks as a guide, relevant variables can be identified from NHANES. These should be variables that are assumed to have some predictive power for expected task performance and are therefore useful for defining meaningful user strata. Strata are identified via statistical clustering. In this paper we discuss how this could be applied to a selected activity—the fine hand manipulation task (pressing buttons, turning knobs, inserting plugs, etc.).

Variables from the 2017–2018 NHANES survey were evaluated for inclusion. An important step in using a database to model user population strata is consulting with key stakeholders to determine what tasks are considered "critical" and what skills are required to accomplish the tasks. Given the focus of the selected case study activity, primarily physical variables were included as directly relevant to fine hand manipulation. Specific diseases were justified using the International Classification of Functioning, Disability, and Health (ICF) core sets. ICF core sets link standardized terminology for human functioning and health developed by the ICF to specific disease categories [56]. If a core set associated with a disease contained the variable "Fine hand use", the disease was included. Other variables were justified using past findings. Age was also included as a sociodemographic variable because of the relationship between age and reduced mobility. The variables included, the format of the variable, and their justification for inclusion are listed in Table 3.

**Table 3.** Name, format, and justification for each variable included in the recruitment strata analysis.

| Variable | Format | Justification |
|---|---|---|
| Age | Continuous | Age and decreased hand mobility are associated [57]. |
| Reported difficulty using fork, knife, or cup | Ordinal—No difficulty, Some difficulty, Much difficulty, Unable to do, Does not do | Activity is a specific case of "fine hand manipulation" |
| Reported difficulty grasping/holding small objects | Ordinal—No difficulty, Some difficulty, Much difficulty, Unable to do, Does not do | Activity is a specific case of "fine hand manipulation" |
| Reported having: <br> - Arthritis <br> - Gout <br> - Bone/joint injury | Binary—Yes or No | "Find hand use" linked with post-acute musculoskeletal disease ICF core set [58]. |
| Reported having: <br> - Congestive heart failure <br> - Angina/angina pectoris | Binary—Yes or No | "Fine hand use" linked with cardiopulmonary post-acute ICF core set [59]. |
| Reported having a stroke | Binary—Yes or No | "Fine hand use" linked with stroke ICF core set [60]. |
| Reported physical activity at work | Ordinal—None, Moderate, Vigorous | Physical activity is associated with fine motor skill [61,62]. |
| Reported physical activity recreationally | Ordinal—None, Moderate, Vigorous | Physical activity is associated with fine motor skill [61,62]. |

These variables are not all encompassing of task performance prediction, but they do represent general population capabilities and provide a clearer picture of user heterogeneity for developing recruitment strata. These variables are in the questionnaire section of the survey, aside from age, which is in the demographics section.

NHANES data is open access and can be downloaded from the NHANES website. For the questionnaire section, data is split into multiple subject-specific files. Responses are deidentified but can be linked by a "Respondent Sequence Number". All the relevant data files were downloaded, and participant sequence numbers were filtered out if they did not report having hypertension. Then, files were linked together to form a dataset for hypertensive respondents. In all, this included a sample of 1628 unique respondents.

To identify recruitment strata, this data was subjected to statistical clustering. While many different algorithms could be used, gaussian mixture model (GMM) clustering is used here. Gaussian mixture models are a model-based clustering algorithm that assumes data are generated from several sub-populations that follow gaussian distributions [63]. Distribution means and variances are fit to the data using the expectation-maximization algorithm. Data points are given "soft assignments" to clusters based on their probability of belonging to each distribution. Further, because the data is of mixed types (continuous, ordinal, and binary), the R package clustMD is used because it is formulated to accept mixed-data [64]. GMM is also preferred here because it provides a statistical framework for evaluating the appropriate number of mixed-data clusters. Bayesian Information Criterion (BIC) can be used to compare the number of clusters and select a number that maximizes model likelihood while penalizing model complexity [64]. While generally more computationally expensive than other clustering algorithms, this was determined not to be an issue because of the relatively small sized dataset [65]. Cluster quantities ranging from 2–10 were evaluated as potential model candidates.

The maximum number of clusters extracted was limited due to the feasibility of meeting stratification goals for each identified cluster in a human performance study.

## 3. Results

Presented in this section are the results of Case Study 1 and Case Study 2.

### 3.1. Case Study 1 Results

Table 4 displays the resulting joint angles for the 95th percentile female hand for the grasping simulation. Table 5 displays the resulting joint angles for the 5th percentile female hand. Total finger deviation is shown for each finger summed across all joints, and total deviation is shown summed across each finger.

**Table 4.** Resulting finger joint deviation from neutral position for the 95th percentile female hand.

| Case | Index Deviation (deg) | Middle Deviation (deg) | Ring Deviation (deg) | Pinky Deviation (deg) | Total Deviation (deg) |
|------|------------------------|-------------------------|----------------------|-----------------------|------------------------|
| 1 | 103.55 | 122.02 | 123.41 | 135.96 | 484.93 |
| 2 | 104.12 | 119.18 | 115.92 | 136.20 | 475.42 |
| 3 | 103.54 | 119.62 | 121.16 | 136.52 | 480.84 |
| 4 | 97.88 | 112.86 | 119.81 | 130.89 | 461.43 |
| 5 | 98.16 | 117.47 | 117.90 | 126.53 | 460.07 |
| 6 | 98.28 | 116.25 | 117.88 | 130.85 | 463.25 |
| 7 | 88.94 | 111.82 | 111.97 | 124.70 | 437.41 |
| 8 | 87.40 | 107.10 | 110.33 | 122.61 | 427.44 |
| 9 | 89.06 | 109.54 | 110.02 | 120.69 | 429.31 |

**Table 5.** Resulting finger joint deviation from neutral position for the 5th percentile female hand.

| Case | Index Deviation (deg) | Middle Deviation (deg) | Ring Deviation (deg) | Pinky Deviation (deg) | Total Deviation (deg) |
|------|------------------------|-------------------------|----------------------|-----------------------|------------------------|
| 1 | 89.67 | 116.06 | 119.48 | 133.92 | 459.13 |
| 2 | 89.24 | 110.92 | 105.08 | 134.06 | 439.30 |
| 3 | 88.52 | 111.27 | 115.25 | 134.11 | 449.15 |
| 4 | 80.91 | 103.28 | 113.94 | 126.29 | 424.43 |
| 5 | 84.55 | 110.25 | 112.39 | 120.38 | 427.58 |
| 6 | 81.16 | 106.32 | 112.34 | 126.28 | 426.11 |
| 7 | 69.87 | 100.20 | 104.77 | 117.90 | 392.74 |
| 8 | 69.72 | 97.37 | 104.76 | 117.92 | 389.78 |
| 9 | 68.76 | 97.98 | 100.03 | 112.19 | 378.96 |

It may be useful to visualize the performance variables, as shown in Figure 3. Total finger deviation is plotted for the 5th and 95th percentiles hands.

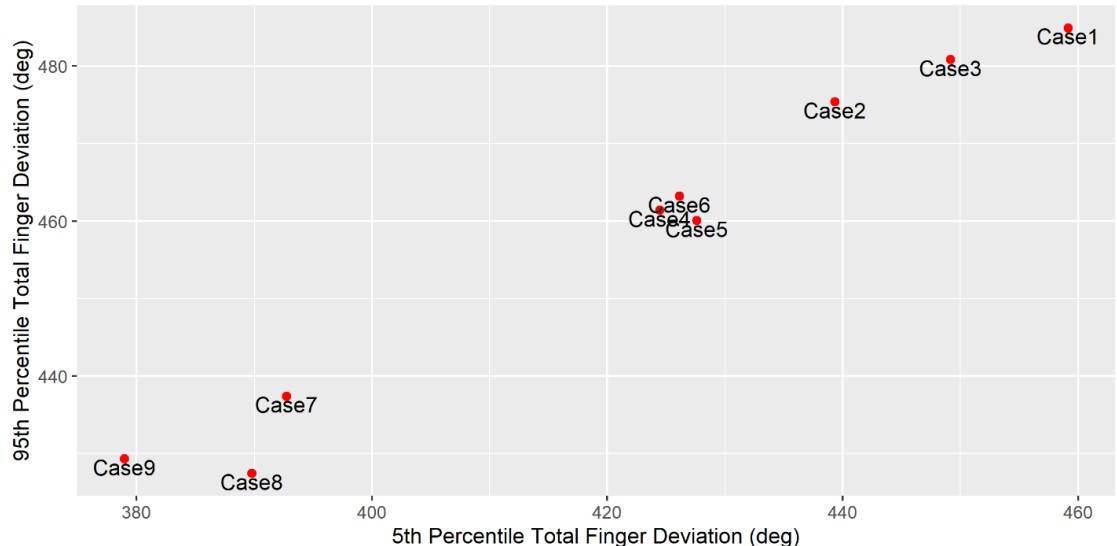

**Figure 3.** Total finger deviation for 5th and 95th percentile hands by case.

Linear model effects are listed in Table 6. Adjusted $R^2 = 0.949$. Estimates are interpreted as change in joint deviation per finger per response variable category.

**Table 6.** Linear model effects for finger joint deviation.

|  | Estimate (deg) | Std. Error | *p*-Value |
|---|---|---|---|
| *Intercept* | 91.283 | 1.279 | <0.001 |
| Case Length (ref. = 10.5 cm) |  |  |  |
| 11.0 cm | −1.689 | 1.044 | 0.111 |
| 11.5 cm | −1.353 | 1.044 | 0.200 |
| Case Breadth (ref. = 3.5 cm) |  |  |  |
| 4.5 cm | −5.247 | 1.044 | <0.001 |
| 5.5 cm | −13.880 | 1.044 | <0.001 |
| 95th Percentile Hand (ref. = 5th Percentile) | 9.250 | 0.852 | <0.001 |
| Finger (ref. = Index) |  |  |  |
| *Middle* | 22.01 | 1.201 | <0.001 |
| *Ring* | 24.62 | 1.201 | <0.001 |
| *Pinky* | 38.59 | 1.201 | <0.001 |

*3.2. Case Study 2 Results*

Presented here are the results of the statistical clustering of selected NHANES variables for the hypertensive population. The optimal number of clusters was determined to be three based on producing the best BIC value. Cluster summaries are shown in Table 7.

**Table 7.** Resulting user group clusters for manipulation tasks in blood pressure monitor use.

| Cluster | | 1 | 2 | 3 |
|---|---|---|---|---|
| n (%) | | 913 (56.1) | 334 (20.5) | 381 (23.4) |
| Median Age | | 70 | 52 | 70 |
| Difficulty using fork, knife, cup (%) | No difficulty | 100 | 97.6 | 65.9 |
| | Some difficulty | 0 | 1.8 | 28.6 |
| | Much difficulty | 0 | 0.6 | 4.7 |
| | Unable to do | 0 | 0 | 0.3 |
| | Does not do | 0 | 0 | 0.5 |
| Difficulty grasp/holding small objects (%) | No difficulty | 90.7 | 80.5 | 15.2 |
| | Some difficulty | 4.8 | 17.7 | 62.5 |
| | Much difficulty | 3.7 | 1.5 | 17.3 |
| | Unable to do | 0.8 | 0.3 | 4.7 |
| | Does not do | 0 | 0 | 0.3 |
| Reported having ... | | | | |
| Arthritis (%) | - | 52.1 | 45.8 | 84.0 |
| Gout (%) | - | 11.6 | 9.0 | 17.3 |
| Bone/joint injury (%) | - | 8.9 | 24.6 | 27.6 |
| Congestive heart failure (%) | - | 6.0 | 3.0 | 24.4 |
| Angina/angina pectoris (%) | - | 5.4 | 2.7 | 17.6 |
| Stroke (%) | - | 8.0 | 4.2 | 28.9 |
| Physical work activities (%) | None | 67.8 | 23.7 | 70.1 |
| | Moderate | 23.1 | 21.9 | 19.1 |
| | Vigorous | 9.1 | 54.5 | 10.8 |
| Physical recreational activities (%) | None | 61.7 | 47.9 | 86.6 |
| | Moderate | 31.5 | 26.6 | 11.8 |
| | Vigorous | 6.8 | 25.4 | 1.6 |

## 4. Discussion

In this paper, the value of modeling the user population prior to engaging with users for a design validation or human performance study was discussed. The value of these models increases when working with highly heterogeneous or difficult-to-access populations. Two case studies were presented that can be thought of as sequential, where the methods demonstrated in Case Study 1 are used to reduce the number of design parameters for a subsequent live-participant study. The methods demonstrated in Case Study 2 are then used to develop a protocol for stratification and recruitment of participants. Both efforts seek to reduce the burden of performing such a study. First, the results and implications of Case Study 1 are discussed.

### 4.1. Benefits of Using DHM to Reduce Product Parameters

DHMs are useful for reducing product parameters in formative design analysis because of the low resource burden required for their application. While potentially time intensive without prior experience, the costs associated with DHM use are only those to procure the software [66]. This is ideal in early design stages when many candidate design solutions must be explored, such as with the handheld device study casing parameters. Though performance measures are limited with DHM use, evaluating 9 separate cases using conventional methods with live subjects may have been costly and

time-consuming. DHMs allow designers to at least narrow down candidate solutions to a feasible number for further testing.

### 4.2. DHM Insights from the Case Study

In Figure 3, 3 groups of cases can be seen corresponding to each case breadth (3.5 cm, 4.5 cm, 5.5 cm). As one might anticipate, 5.5 cm led to the smallest joint deviations (largest grip diameter), and 3.5 cm led to the largest joint deviations. This follows an expected linear trend and was verified with the regression analysis. Within each group, there are relatively smaller differences in joint deviations and there is no apparent trend resulting from case length. This is reflected in the regression analysis as well, which did not return any significant association between case length and joint deviation. The differences in joint deviation within each case of the same breadth may be due to noise. DHMs can produce small variations in results due to the large number of ways a human model can be positioned, especially for a hand model with many degrees of freedom. DHM accuracy is highly dependent on the modeler [66]. Overall, however, the model $R^2$ value indicated that ~95% of joint variability was accounted for by model estimates, indicating that only a small amount of variability can be attributed to noise.

Based on these results, one might select to eliminate cases 4–9 from future studies. Overall, these cases performed worse than the cases in the 3.5 cm width group based on joint deviation. Alternatively, one could select a case from each of the 4.5 and 5.5 cm width groups to include in further experimental testing to facilitate model validation. Model validation is crucial to ensure human movement is replicated as closely as possible [67]. Selecting one case from each group ensures model validation is comprehensive without requiring all cases to be included. While case 1 was determined to be "optimal" given the selection criteria, there was not enough evidence to demonstrate case length had a significant influence on joint deviation. Therefore, it may be worthwhile to further explore all the cases in the 3.5 cm width group.

### 4.3. DHM Applications and Limitations

When using DHMs to evaluate design parameters and the interaction is static or relatively simple, it is suggested some amount of automation be used to expedite model throughput. Past work has demonstrated this to be a viable option [68]. Based on the results from Case Study 1, it is believed these simulations can be reliably and consistently automated. Doing so encourages standardized results and opens the opportunity to evaluate more design parameters. For Case Study 1, there were 18 total simulations that needed to be performed (9 cases * 2 hand sizes). Even for this relatively simple case study, performing all these simulations completely manually could be quite time consuming. For an interaction where many human parameters must be varied or there are many candidate design parameters, this could result in an infeasible number of simulations to perform manually. Further, even when simulations are automated, one should consider the additional time required to validate model results. A robust follow-up study should include some of the "eliminated" parameters to validate model results, and this will add additional time.

This case study suggests that DHM software can be useful for modeling simple interactions. Examples of other interactions that could be modeled are static poses, such as when evaluating wearables or spacing in seated environments or vehicle cabins. Simple interactions involving few joints, such as turning of the head or reaching for an object, may also be reliably modeled. These results suggest that there may be difficulties when simulating more complex and dynamic movements with DHM software. There was some variability in the case study results that could not be accounted-for due to varied human or design parameters. Automating the scaling and posing of hand models meant some quality assurance was lost in the process because simulations were not directly observed. Further, the initial positioning of the hand and the application of constraints is subject to some variability, even if the operator takes great care to assure consistency. It is believed that, with more complex interactions, there would be more opportunities for this type of variability to occur. Therefore, in these cases, it may

be best to manually scale and pose the human models for a smaller number of simulations. This may limit the number of parameters that can be evaluated, depending on resources.

In general, incorporating more complex biomechanical models into theses primarily kinematic models such that the simplicity of implementation is not lost would be beneficial. The inverse kinematic techniques that are used in Jack are not integrated with musculoskeletal models, which can lead to unpredictable results. Inverse kinematics seek to find joint angles for a given end effector position. There may be many candidate solutions for this, which the software cannot judge for realism.

An additional limitation when using DHMs to model human variability is that the default or underlying models and data may not be applicable to the subject user population [69]. For example, the default anthropometric data used to scale the models may not appropriately replicate the subject population anthropometry. In Case Study 1, hand dimensions were selected to simulate a non-general population. In actual practice, these dimensions should be based off existing statistics. DHMs cannot indicate the source of human performance variability; the practitioner must still make the decision on what and to what extent human parameters must be varied in the simulation. DHMs also do not provide any information about the distribution of characteristics leading to performance variability in hard-to-access and rarely studied user populations. Anthropometric distributions, musculoskeletal and strength calculations, workload predictions, etc. are all based on existing data and models may not be tailored to the population of interest. Tailoring these models for use in non-general populations may be burdensome or impossible given the expertise of the person applying them.

### 4.4. Benefits of Using Existing Data to Identify Recruitment Strata

In this paper, we proposed the use of existing datasets to identify groups of users with similar characteristics and capabilities. While clustering users by characteristics related to product interaction isn't completely novel [45], this work demonstrates how the concept can be used for specialty populations by utilizing data in a novel way. User group identification was demonstrated on the hypertensive medical device user population, where NHANES data was statistically clustered. The results were dense representations of similar users. These can be used to derive user personas and, using the group proportions identified, goals can be set for representation in recruitment for a subsequent study. This is useful because it allows primary user strata to be identified without directly needing to engage the population. For difficult-to-access populations, such as medical device users, this technique can provide a cost-effective approximation of population heterogeneity.

These strata could be useful for recruitment to ensure population representation. Pre-study questionnaires could be used to find participants that conform to strata and determine inclusion/exclusion of participants. These questionnaires can also be used to provide real-time data on strata goal achievement, potentially leading to a re-calibration of recruitment efforts at different points in the study (i.e., focusing recruitment on an underrepresented strata). Similarly, if a study has approval for access to medical records through collaboration with a medical facility, the records could be used to compile a strategic recruitment list of eligible subjects that are likely to fall within a specific stratum. If the study is already completed, post-study questionnaires could be used to evaluate how well population strata were represented.

### 4.5. Recruitment Strata Case Study Insights

Based on the cluster analysis, 3 dominant groups emerged. Cluster 1 is the most predominant group and is primarily comprised of elderly individuals who don't report difficulties with physical independence. The majority do report arthritis though, and primarily lead a sedentary lifestyle. Cluster 2 is a younger group of users who primarily do not report any difficulties with physical independence or diseases that could interfere with manipulation tasks. This group also reports weekly vigorous work activity, and at least weekly moderate recreational activity. Cluster 3 is again a group of elderly users. Most of them report difficulty performing grasping tasks, arthritis is highly prevalent, and they generally lead sedentary lifestyles. It is suspected that Cluster 2 would be

the top task performers, Cluster 3 would be the bottom task performers, and Cluster 1 would sit somewhere in-between.

Of the comorbidities included in the clustering, only arthritis presented itself as predominant in any group. With the inclusion of more user clusters, a different comorbidity might emerge as predominant for a group. It is important to remember, however, that the purpose of clustering is to balance user representation with the practical cost of representing them in a study by finding common user characteristics that are densely represented in the population. In an ideal scenario, every combination of relevant user characteristic would be represented with a sizeable sample population, but the feasibility of this is obviously limited. Even with 3 clusters, adequate recruitment could be difficult depending on the availability of those populations. Representing heterogenous user populations in live-participant studies will always be more difficult than recruiting blindly as if the population is homogenous. This work seeks to address the trade-off between minimizing cost and representing more users in a study, however the researcher performing the study must ultimately judge where they fall between those two objectives.

*4.6. Recruitment Strata Applications and Limitations*

It is reasonable to ask why recruitment strata for usability studies should not be defined using demographic or geographic variables. In general, these variables have tenuous causal relationships to task performance. The elderly may be more likely to perform poorly at visual tasks, but this is because they are more likely to have some sensory or cognitive impairment, not directly because they are elderly. While more convenient because of their wide availability, other characteristics directly linked to performance would likely be better suited for identifying homogenous groups with respect to task performance. Doing so is important because it helps to ensure that certain types of users are not over or underrepresented in a study, which can be costly for manufacturers if a study is determined to have produced poor results and must be repeated. It can also be costly for users if poor results are undiscovered or ignored and their usability needs go unaddressed.

Future work should seek to validate these strata as useful tools in recruitment. Demonstrating that study samples recruited using these strata produce significantly different results than a study utilizing differently, or non-stratified sampling could validate this as a useful process. Additional future work could include utilizing this process on different datasets and for different domains. Healthcare has many disease specific user populations that differ from the general population, but so do many other domains. The military population, for example, is generally considered distinct from the general population [43] and might benefit from using this type of stratification when quantifying human performance.

**5. Conclusions**

In this paper, two techniques were discussed for reducing the complexity of a study seeking to quantify human performance in product interaction. This is critical when engaging highly heterogeneous, difficult-to-access populations to minimize the cost and difficulty of the study. The first technique focused on using DHM to reduce the number of design parameters to include in a study. A case study demonstrated how DHM software could be used to evaluate a set of design parameters. The results of the case study demonstrated that DHMs can be used to eliminate design parameters from consideration for future studies. The second technique focused on identifying meaningful user strata based on characteristics that influence human interaction with a system. A case study was performed examining the hypertensive population and interaction with blood pressure monitors. Relevant variables were identified from NHANES and used to cluster users into groups. These groups can be useful for recruiting subjects in a study such that the capabilities of the user population are adequately represented. The needs of highly heterogeneous user populations with characteristics that differ from the general population often go overlooked in the product design process. Finding ways to

ease the burden of engaging with these populations for manufacturers is important to ensure design of safe and efficient systems.

**Author Contributions:** Conceptualization, B.M.K. and M.V.-C.; methodology, B.M.K. and M.V.-C.; software, B.M.K.; validation, B.M.K. and M.V.-C.; formal analysis, B.M.K.; investigation, B.M.K.; resources, M.V.-C.; data curation, B.M.K.; writing—original draft preparation, B.M.K.; writing—review and editing, B.M.K. and M.V.-C.; visualization, B.M.K.; supervision, M.V.-C.; project administration, B.M.K. and M.V.-C. All authors have read and agreed to the published version of the manuscript.

**Funding:** This research received no external funding.

**Acknowledgments:** We would like to thank the University of Maryland undergraduate research assistants at the Hybrid-Simulation and Integration Lab who assisted with simulation development and data collection.

**Conflicts of Interest:** The authors declare no conflict of interest.

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
