# Peer review of "Virtual Modeling of User Populations and Formative Design Parameters"

_systems, doi:10.3390/systems8040035_

Round 1

Reviewer 1 Report

Dear Authors,

I appreciate the efforts that went into this paper and I appreciate the opportunity to review a paper which considers the aspects related usability. This paper "Virtual Modeling of User Populations and Formative Design Parameters" cover a comparative analysis of different models to apply in design. On the other hand, the paper needed modifications and I would like to give more specific recommendations for the authors' that I believe will improve the paper. I think this paper will fit well in the journal after the revisions.

MATERIALS AND METHODS

2.1 Case Study 1: Using DHM to Reduce Product Parameters

Handheld medical devices is a wide variety of systems which different uses. Some concepts related to usability are mentioned in several standards such as IEC 62366:2007, IEC 62366:2007+A1:2014, IEC 62366-1:2015, which define Usability as characteristic of the user interface that establishes effectiveness, efficiency, and user satisfaction. Furthermore, In order to minimize use errors and use associated risks, reasonable Usability must be achieved by using usability Engineering process. (IEC 62366:2007 and A1:2014). I recommend to specify the usability characteristics regarding the suggested handheld and the mentioned standards.

2.2. Case Study 2: Using Existing Data to Identify Recruitment Strata

Sample of the database are not included. The sample should include a statistical justification.

Line 341: Variables and the data type indicated are not explained. “Age” is briefly commented, but there is no reference included. I recommended to review previous studies related all the possible variables and also, indicated exclusion variables.

RESULTS

The analysis of the case 1 and 2 are not linked and related. I recommend trying to connect both cases in order to offer more information about the traceability of the results.

Case 2 statistics do not consider reliability coefficients (i.e. Cronbach’s alpha) and others coefficients such as Pearson, Kaiser-Meyer Olkin, Barlett’s test or others… I recommend providing coefficients of statistical analysis.

DISCUSSION AND CONCLUSION

I have not identified any citation and reference in these parts. I suggest including some citation of articles/papers according to the analysis.

REFERENCES

I recommend adding the DOI in articles and webpage in the rest of the cases.

Reviewer 2 Report

While the accuracy of simulation using DHMs is critical, there seem many assumptions involved in the simulation model.

Some of them may be;
1. Each finger equally contributes to gripping comfort (?).
2. Joint angle deviation is related to gripping comfort (?).
3. When does the hand model stop flexing fingers? How hard should finger skin be pressed?
4. Etc.?

The authors need to clearly describe any assumptions made for the simulation model and discuss their impacts on the results of their study.

Line 102
An acronym (SES) requires its full name when it is first used.

Line 160
What are these characteristics?

Figure 1
As the hand-held object is not rectangular, two dimensions of device width and length seem insufficient to characterize this object.

Line 282
Define the neutral position. Describe how each joint angle is defined preferably using figures.

L442
The authors need to explain why cases 4-9 should be eliminated.

L454
The authors need to compare the required times for a hand simulation-based study and a live subjects-involved study.
I think a hand simulation-based study is also time-consuming and requires simulation validation.

Line 472-474.
The sentence should be grammatically checked.

Line 510
Is it practically possible to recruit people representative of three clusters? It would be more difficult to recruit people from these specific clusters.

Tables 2 and 3
Why are thumb angles excluded?

Table 5
Are there any differences in the hand anthropometric data and the neutral position between hypertensive and non-hypertensive populations? I don't see any links between DHM and three identified clusters.

Round 2

Reviewer 1 Report

The authors have made the changes and these changes have been justified.

For this reason, I consider that the document is ready to be published.

Reviewer 2 Report

The authors have successfully addressed my previous comments.